# RNA binding to human METTL3-METTL14 restricts $N^6$-deoxyadenosine methylation of DNA in vitro

Shan Qi[1,2†], Javier Mota[1†], Siu-Hong Chan[3], Johanna Villarreal[1], Nan Dai[3], Shailee Arya[1], Robert A Hromas[4], Manjeet K Rao[1], Ivan R Corrêa Jr[3], Yogesh K Gupta[1,2]*

[1]Greehey Children's Cancer Research Institute, University of Texas Health at San Antonio, San Antonio, United States; [2]Department of Biochemistry and Structural Biology, University of Texas Health at San Antonio, San Antonio, United States; [3]New England Biolabs, Ipswich, United States; [4]Division of Hematology and Medical Oncology, Department of Medicine, University of Texas Health at San Antonio, San Antonio, United States

*For correspondence:
guptay@uthscsa.edu

†These authors contributed equally to this work

**Abstract** Methyltransferase like-3 (METTL3) and METTL14 complex transfers a methyl group from $S$-adenosyl-L-methionine to $N^6$ amino group of adenosine bases in RNA (m6A) and DNA (m6dA). Emerging evidence highlights a role of METTL3-METTL14 in the chromatin context, especially in processes where DNA and RNA are held in close proximity. However, a mechanistic framework about specificity for substrate RNA/DNA and their interrelationship remain unclear. By systematically studying methylation activity and binding affinity to a number of DNA and RNA oligos with different propensities to form inter- or intra-molecular duplexes or single-stranded molecules in vitro, we uncover an inverse relationship for substrate binding and methylation and show that METTL3-METTL14 preferentially catalyzes the formation of m6dA in single-stranded DNA (ssDNA), despite weaker binding affinity to DNA. In contrast, it binds structured RNAs with high affinity, but methylates the target adenosine in RNA (m6A) much less efficiently than it does in ssDNA. We also show that METTL3-METTL14-mediated methylation of DNA is largely restricted by structured RNA elements prevalent in long noncoding and other cellular RNAs.

## Editor's evaluation

This manuscript will be of interest to researchers in the fields of nucleic acid chemical biology in general and diseases related to nucleic acid methylation in particular. The data presented support the conclusions of the paper within the current context, provide new evidence and plausible explanations to previously inexplicable mechanisms.

## Introduction

$N^6$-methyladenosine (m6A) is considered a major covalent modification of the adenosine (A) base in coding and noncoding (nc) RNAs (*Desrosiers et al., 1974*; *Bokar et al., 1994*; *Rottman et al., 1994*; *Dominissini et al., 2012*; *Meyer et al., 2012*). It is linked to diverse physiological processes, including – but not limited to – RNA turnover (*Ke et al., 2017*; *Knuckles et al., 2017*; *Liu et al., 2020*), stem cell differentiation (*Batista et al., 2014*; *Geula et al., 2015*,) oncogenic translation (*Lin et al., 2016*; *Barbieri et al., 2017*; *Vu et al., 2017*; *Choe et al., 2018*), and DNA damage repair (*Xiang et al., 2017*; *Zhang et al., 2020*). In human cells, most m6A in cellular RNAs is installed by

methyltransferase like-3 (METTL3) and METTL14 methyltransferases (*Bokar et al., 1994*; *Dominissini et al., 2012*; *Liu et al., 2014*), both of which belong to the β-class of *S*-adenosyl methionine (SAM)-dependent methyltransferases ($N^6$-MTases) (*Bujnicki et al., 2002*; *Woodcock et al., 2020*) that also include $N^6$-deoxyadenosine DNA methyltransferases (m⁶dA MTases), especially those from Type III restriction-modification (R-M) systems in bacteria (e.g., Mod$_{A/B}$ dimer of EcoP15I) (*Gupta et al., 2015*). The first structures of METTL3-METTL14 complexes suggested common features in EcoP15I and METTL3-METTL14 (*Wang et al., 2016b*; *Sledz and Jinek, 2016*). We analyzed these MTases and found high structural similarity of the cores of human METTL3, METTL14, and EcoP15I, suggesting their evolutionary origin from a common ancestor (*Figure 1a–e* and *Figure 1—figure supplement 1*).

While m⁶dA in bacterial DNA is deposited in a strictly sequence-dependent manner in double-stranded DNA (dsDNA) (*Gupta et al., 2015*; *Malone et al., 1995*), m⁶A in RNA by human METTL3-METTL14 shows less stringent sequence dependency for sequences flanking the target **A** within the recognition motif DR**A**CH (D = A/G/U, *R* = A/G, H = A/C/U) (*Bokar et al., 1994*; *Liu et al., 2014*; *Linder et al., 2015*). Such relaxed specificity maybe required to expand the repertoire of target transcripts of METTL3-METTL14. However, this does not correlate with low levels of m⁶A in cellular RNA, which remains at ~0.1% of total pool of ribonucleotides in cultured mammalian cells (*Dubin and Taylor, 1975*). Of note, recent evidence suggests that all m⁶A is deposited co-transcriptionally, with most occurring on chromatin-associated RNAs, for example, nascent pre-mRNA (CA-RNA), promoter-associated RNA, enhancer RNA, and repeat RNA elements (*Ke et al., 2017*; *Knuckles et al., 2017*; *Liu et al., 2020*; *Slobodin et al., 2017*). Consistently, the METTL3-METTL14 m⁶A writers can be recruited to chromatin by modified histone tails (e.g., H3K36me3) (*Huang et al., 2019b*) or the transcription machinery (*Barbieri et al., 2017*; *Slobodin et al., 2017*). Certain long noncoding (lnc) RNAs that are essential for regulated transcription, such as chromatin-associated *XIST* (*Patil et al., 2016*) and *NEAT2* (*MALAT1*) (*Liu et al., 2015*), also harbor m⁶A marks. A role of m⁶A in DNA damage response and repair has also been suggested (*Zhang et al., 2020*; *Xiang et al., 2017*). In those contexts, METTL3 localizes at sites of dsDNA breaks and installs the m⁶A on DNA-damage-associated RNAs (*Zhang et al., 2020*). METTL3 is also emerged as a pro-viral host factor in SARS-CoV-2 pathogenesis. Consistently, genetic depletion or inhibition of METTL3 by small molecules triggers innate immune response and limits viral growth (*Burgess et al., 2021*; *Li et al., 2021*).

The increasingly appreciated roles of m⁶A in nucleic acid metabolism thus seem to converge to a common point: the occurrence of m⁶A at chromatin, a predominantly DNA- and RNA-rich environment inside the nucleus. A recent study showed that METTL3-METTL14 can convert dA to m⁶dA within equivalent DR**A**CH (D = A/G/U, *R* = A/G, H = A/C/U) motif in a single-stranded DNA (ssDNA) with a 2.5-fold preference over the equivalent RNA in vitro (*Woodcock et al., 2019*). Since m⁶dA levels in mammalian DNA are extremely low (0.006–0.01%) and thought to originate from RNA catabolism rather than from SAM-dependent MTase reactions (*Musheev et al., 2020*), fundamental questions about substrate(s) specificity and mechanisms of action of METTL3-METTL14 have emerged. By using classical biochemical and mass spectrometry assays, we show that METTL3-METTL14 preferentially methylates N⁶dA on ssDNAs to m⁶dA, and this activity is largely regulated by structured RNA elements prevalent in lnc RNAs, that are also found in other cellular RNAs. Thus, our work provides a framework to explore a new regulatory axis of RNA-mediated restriction of $N^6$-deoxyadenosine (m⁶dA) methylation in mammalian genomes.

## Results

We compared the MTase cores of (Mod$_B$) of EcoP15I (aa 90–132, 169–261, 385–511, PDB: 4ZCF *Gupta et al., 2015*) and METTL14 (aa 116–402, PDB: 5IL0 *Wang et al., 2016b*). A secondary structure-based superposition (*Krissinel and Henrick, 2004*) of these two structures revealed common features and similar arrangement of canonical MTase motifs (motifs I and IV-X), including the motif IV (D/EPPY/W/L) that surrounds the Watson-Crick edge of the flipped target adenine base (*Figure 1a–e* and *Figure 1—figure supplement 1*). Previous studies identified three loops encompassing the sequence intervening MTase motifs IV and V (loop 1), VIII and VIII' (loop 2), and motifs IX and X (loop 3) contribute to substrate binding (*Wang et al., 2016b*; *Sledz and Jinek, 2016*; *Wang et al., 2016a*). Due to lack of experimental structures of METTL3 or Mod$_A$ (in complex with a flipped adenine base), we chose Mod$_B$ and METTL14 for this comparison. While the MTase domain of METTL14 is enzymatically inactive, it harbors features that contribute to substrate binding (*Wang et al., 2016b*; *Sledz*

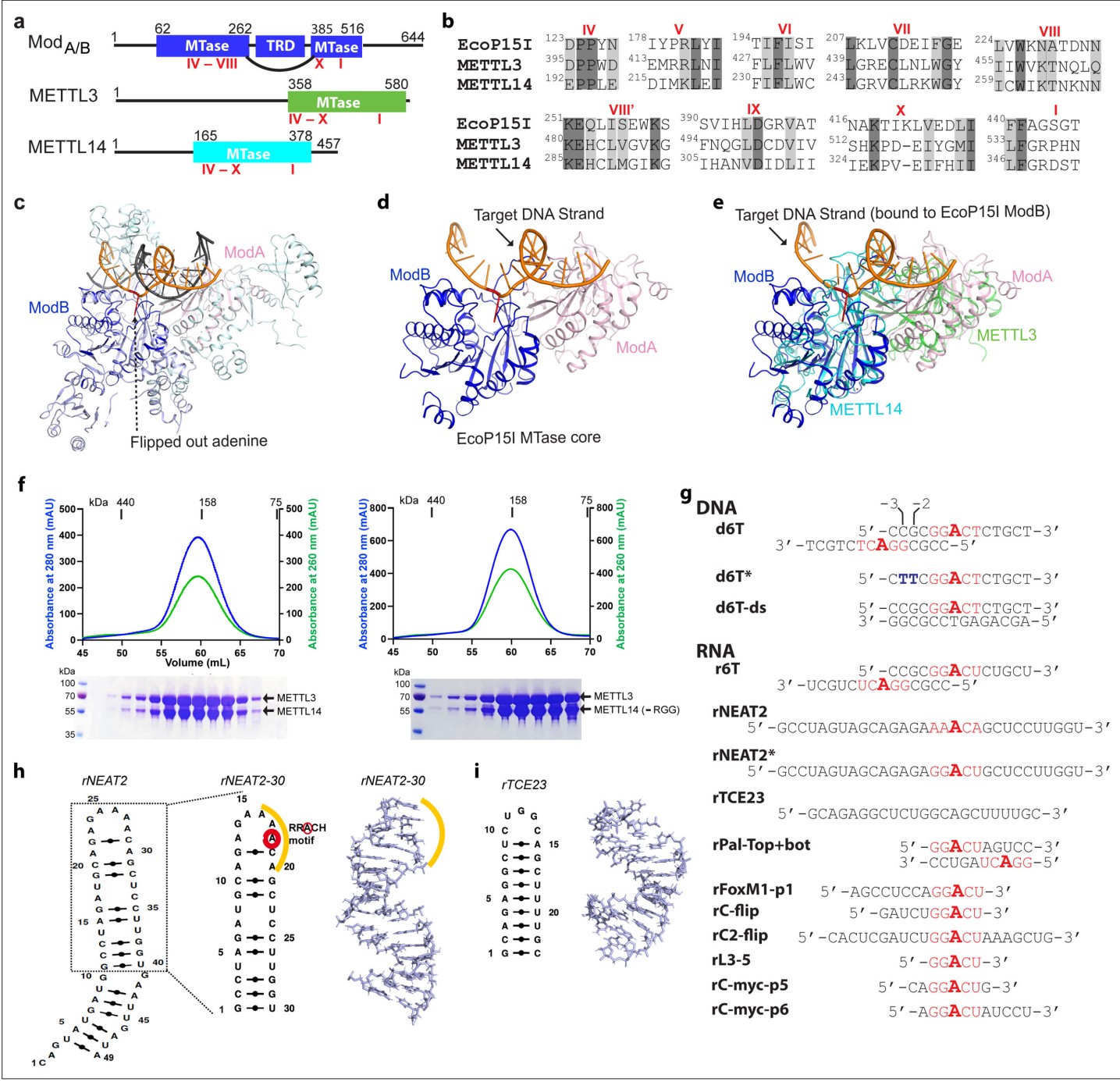

**Figure 1.** Structural similarity, purification of methyltransferases, and substrate designing. (**a–b**) Domain architecture of Mod subunit of EcoP15I, human methyltransferase like-3 (METTL3), and human METTL14 methyltransferases (MTases). All three members belong to the β-class of SAM-dependent MTases and exhibit a sequential arrangement of motifs (IV-X followed by motif I) (***Bujnicki et al., 2002***; ***Woodcock et al., 2020***; ***Malone et al., 1995***). Motif IV (D/EPPY/W/L) and I are associated with the recognition of target adenine base (***Gupta et al., 2015***) and co-factor (SAM) binding (***Wang et al., 2016b***; ***Sledz and Jinek, 2016***; ***Wang et al., 2016a***), respectively. (**c**) Crystal structure of EcoP15I-DNA complex (PDB ID: 4ZCF) (***Gupta et al., 2015***): Mod$_A$ MTase (cyan), Mod$_B$ MTase (blue), non-methylating DNA strand (gray), target (methylating) DNA strand (orange), flipped adenine base (red stick). The methyltransferase cores of Mod$_B$ and Mod$_A$ are shown in dark blue and light pink, respectively. The Res subunit of EcoP15I was omitted for clarity. (**d**) MTase domains of two Mod subunits (ModA/B) of EcoP15I with target DNA strand. The regions flanking the methyltransferase core (e.g., CTD and TRD) are omitted for clarity. Only the region encompassing the MTase core (aa 90–132, 169–261, and 385–511) of EcoP15I Mod was selected for the alignment with the methyltransferase core of METTL3 (aa 358–580) and METTL14 (aa 165–378). (**e**) An overlay of MTase domains of EcoP15I and METTL14 (PDB ID: 5IL0) shows structural similarity within MTase folds (rmsd = 2.55 Å over 278 Cα atoms). RNA strand here was modeled based on the respective methylating strand in the EcoP15I structure.(**f**) Chromatogram of final size exclusion chromatography (SEC) step of purification showing

*Figure 1 continued on next page*

*Figure 1 continued*

METTL3-METTL14 complexes (left, full-length; right, METL3-METTL14$_{[-RGG]}$) co-eluted as single homogenous species. Blue, absorbance at 280 nm; green, absorbance at 260 nm (A260). Coomassie stained gels (lower panels) confirm high purity of METTL3-METTL14 proteins in the SEC peak fractions. (**g**) Sequence of DNA and RNA oligonucleotides used in this study. All oligos have a covalently attached 5′-fluorescein (not shown). (**h**) Secondary structure of rNEAT2 and its 3D model as predicted by MC-SYM (*Parisien and Major, 2008*). rNEAT2 harbors a potential DRACH motif (yellow line). (**i**) Secondary structure of rTCE23 RNA and its solution NMR structure (PDB ID: 2ES5) (*Oberstrass et al., 2006*).

The online version of this article includes the following figure supplement(s) for figure 1:

**Figure supplement 1.** Potential mode of DNA recognition.

*and Jinek, 2016*; *Wang et al., 2016a*). These observations prompted us to address the prevailing question about substrate specificity of the human full-length METTL3-METTL14 enzyme complex. We co-purified the full-length human METTL3-METTL14 complex and its truncated form devoid of the RGG motif of METTL14 (METL3-METL14$_{-RGG}$) from insect cells (see details in Materials and methods section) (*Figure 1f*). The RGG motifs are clustered sequences of arginines and glycines. These motifs are commonly present in diverse set of RNA-binding proteins that play important roles in physiological processes, such as RNA synthesis and processing (*Thandapani et al., 2013*). In human METTL14, a total of seven RGG triplets and two RG motifs are present at its C-terminus tail (aa 408–457). The region also harbors a number of aromatic amino acids, which can further stabilize the bound nucleic acids via hydrophobic or stacking interactions. Consistently, this region contributes to substrate binding and m⁶A activity of the METTL3-METTL14 enzyme complex (*Schöller et al., 2018*). The sequence encompassing the RGG motifs in METTL14 is well conserved in higher vertebrates (*Liu et al., 2014*).

Next, we designed RNA substrates of varied lengths (5–30 nucleotides) with at least one canonical m⁶A signature motif (RR**A**CH), including an RNA oligo (rPal-Top+ bat) in which two RR**A**CH sites were arranged in a palindromic fashion (*Figure 1g*). We included a 14-mer DNA oligo (d6T) according to *Woodcock et al., 2019*, that reported that METTL3-METTL14 exhibited highest methyltransferase (MTase) activity on d6T. We also included two structured RNAs with propensity to form a perfect stem-loop (23-mer rTCE23 or also known as SRE; *Oberstrass et al., 2006*) without a DRACH motif, and a 30-mer rNEAT2 (or MALAT1) encompassing one DRACH motif and predicted to form a stem-loop with a bulged stem (*Figure 1h–i*).

The m⁶A mark in mammalian RNAs is widespread and expands beyond internal adenines in mRNAs to lncRNAs (*XIST Patil et al., 2016*, NEAT2 or MALAT1 *Liu et al., 2015*), pri-miRNAs (*Alarcón et al., 2015*), and snoRNAs (*Linder et al., 2015*) – all of which are crucial to gene regulation and maintenance of cellular homeostasis. Of note, *NEAT2* lncRNA can relocate a large subset of transcription units across cellular compartments from polycomb bodies to interchromatin granules (*Yang et al., 2011*). According to this model, a 49-nt long fragment of *NEAT2* (nt 4917–4966) facilitates the recruitment of a mega transcription complex (*Yang et al., 2011*). We observed that this *NEAT2* fragment harbors one RR**A**CH motif (AA**A**CA) in its loop region and fulfils the criteria of m⁶A peak signature (RR**A**CH *Bokar et al., 1994*; *Liu et al., 2014* or DR**A**CH *Linder et al., 2015* or **A**CA *Garcia-Campos et al., 2019*; *Figure 1g*). Thus, we decided to test the activity of METTL3-METTL14 on this regulatory module of the lncRNA. For simplicity of synthesis and purification, we focused on a slightly shorter version (30-nt; *rNEAT2-30*) of *NEAT2-49* in which both the m⁶A signature motif and bulged stem regions were well preserved (*Figure 1g–h*). The rTCE23 could serve as a control – it forms a perfect stem-loop structure in solution – confirmed by solution NMR previously (*Oberstrass et al., 2006*) – but lacks an m⁶A signature motif (*Figure 1g and i*).

We observed higher enzymatic activity of METTL3-METTL14 on the d6T DNA oligo compared with its RNA counterpart (r6T) (*Figure 2a and c*). By closely examining the sequence of the 14-mer d6T/r6T, we found a propensity of these oligos to form duplex DNA (for d6T) or RNA (for r6T): 6 of the 14 nucleotides at 5′-end will pair with a complementary sequence to create a duplex with an 8-nt overhang at the 3′-end of each strand (*Figures 1g and 2d*). As a result of this self-annealing, two adenine bases are available for methylation (m⁶dA) at the end of the short palindromic sequence (5′-CCG•C-GG-3′). We thus hypothesized that d6T (or 6T, as referred to by *Woodcock et al., 2019*) may not truly represent an ssDNA substrate. Thus, we designed a new DNA oligo (d6T*) in which two nucleotides, a deoxycytidine and a deoxyguanosine at –3 and –2 positions (relative to the signature motif RR**A**CH), respectively, were replaced by two thymidines to disrupt base pairing and palindrome formation without affecting the core RR**A**CH motif required for m⁶dA installation (*Figure 1g*). We reasoned

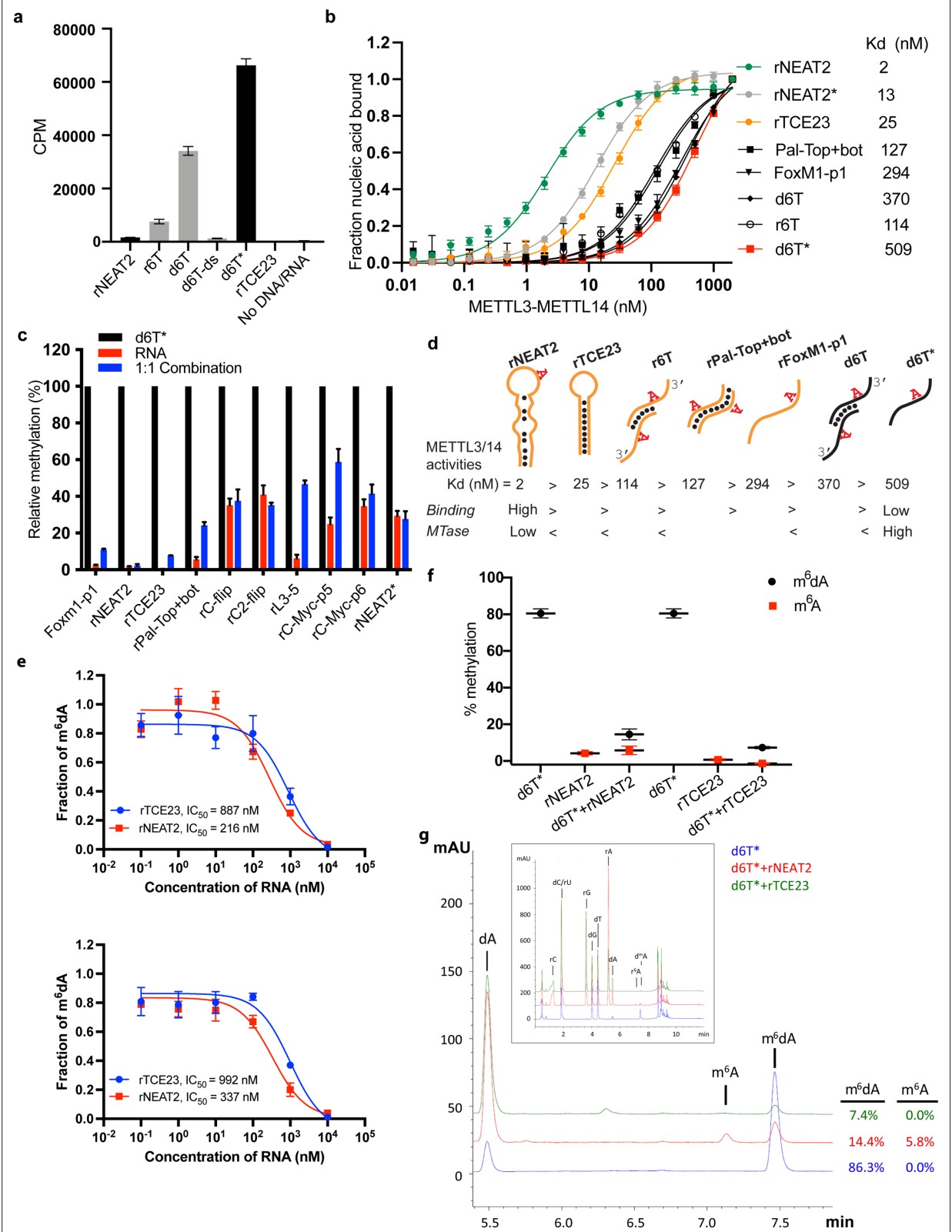

**Figure 2.** RNA-mediated restriction of methyltransferase like-3 (METTL3)-METTL14 activity. (**a**) Methyltransferase activity of METTL3-METTL14 in the presence of various DNA or RNA substrates measured by radiometric assay. CPM, counts per minute. The highest activity was measured with the d6T* oligo. (**b**) FP-based binding assay for DNA and RNA oligos showing highest affinity for rNEAT2 RNA (green) and lowest affinity for d6T* oligo (red). Equilibrium dissociation constants (Kd) for each oligo are shown on the right side of the isotherms. The data were fit into one site-specific binding

*Figure 2 continued on next page*

*Figure 2 continued*

model (Y = Bmax*X/(Kd+ X)). See Materials and methods section and source data for details. (**c**) Methyltransferase activity of METTL3-METTL14 on the respective RNA oligos (red), d6T* DNA alone (black) and equimolar mixture of the two (blue), measured by radiometric assay. (**d**) Predicted secondary structures of each oligonucleotide. Yellow, RNA strand; black, DNA strand. The values of the equilibrium dissociation constants (Kd) shown for each oligonucleotide indicate an inverse relationship between binding affinity and methyltransferase activity. (**e**) Dose-dependent inhibition of METTL3-METTL14 activity by RNA oligos rNEAT2 or rTCE23, as measured by radiometric assay in a reaction buffer containing 5.0 mM NaCl (upper panel) and 50.0 mM NaCl (lower panel). IC$_{50}$, concentration of RNA required to achieve 50% inhibition of the METTL3-METTL14 activity. (**f**) Attenuation of the methyltransferase activity in presence of rNEAT2 or rTCE23, as measured by oligonucleotide intact mass analysis. Quantitation of modified dA (black circle) or rA (red square) is shown in absence or presence of equivalent amounts of rNEAT2 or rTCE23. (**g**) Nucleoside composition analysis of the METTL3-METTL14 reactions. UHPLC chromatograms showing the reaction in absence (blue trace) or in the presence of either rNEAT2 (red trace) or rTCE23 (green trace). The quantitation of the fraction of modified bases in the nucleoside pool was consistent with the results from the oligonucleotide intact mass analysis shown in (**f**). The insert shows the full chromatographic trace with all detected nucleosides. Results presented in panels (**a–c** and **e–f**) are the average of three independent experiments (n = 3) with one standard deviation (s.d.) for each oligonucleotide (shown as error bars). Source data are provided as a Source Data file.

The online version of this article includes the following source data for figure 2:

**Source data 1.** MTase activity on individual RNA/DNA substrates.

**Source data 2.** Binding isotherms of METTL3-METTL14.

**Source data 3.** Comparison of MTase activity on different RNA/DNA substrates.

**Source data 4.** Dose-dependent inhibition of MTase activity by RNA.

**Source data 5.** RNA-mediated attenuation of MTase activity by intact mass analysis.

that if METTL3-METTL14 indeed prefers a ssDNA for methylation, it should show higher activity for d6T*, a true ssDNA substrate, over d6T. Indeed, we found that the methyltransferase activity of METTL3-METTL14 toward d6T* increased by 2-fold compared with d6T (and >12-fold compared with r6T) (*Figure 2a*). Consistent with previous observations (*Woodcock et al., 2019*), METTL3-METTL14 showed no activity in the presence of a perfect duplex version of d6T DNA (d6T-ds). These results confirm a robust activity of METTL3-METTL14 on ssDNA substrates.

We observed no methyltransferase activity on rTCE23 RNA, as expected; but surprisingly, some low (but consistently detectable) methyltransferase activity on *rNEAT2* (*Figure 2a*). We then asked what attributes (sequence, length, and shape) in an RNA substrate are critical for METTL3-METTL14 to yield a high methyltransferase activity. To answer this question, we designed eight RNA oligos with varied length (5- to 22-nt) and sequences (*Figure 1g*). We also varied the position of the RR**A**CH motif and the flanking sequences in these oligos. All oligos comprised one RR**A**CH motif, except the RNA duplex Pal-Top+ bot, where two RR**A**CH sites were arranged in a palindromic fashion within each 10-mer strand. We designed another variant of probe rC2-flip (22-mer, RRACH in the center), rC-flip wherein the RR**A**CH motif resided at the 3'-end. The other three RNA oligos, FoxM1-p1 (13-nt,

**Table 1.** Equilibrium binding constants.

| Nucleic acid substrate | kd in nM (range of Kd in nM) | |
|---|---|---|
| | Full-length METTL3-METTL14 | METTL3-METTL14$_{-RGG}$ |
| **RNA** | | |
| rNEAT2 | 2.1 (1.8–2.3) | 21.2 (19.2–23.5) |
| rNEAT2* | 13.0 (12.3–13.8) | 274.1 (150.7–497.2) |
| rTCE23 | 25.6 (23.2–28.5) | 413 (341–502) |
| rPal-top+ bot | 127 (97.2–167) | 257 (132–491) |
| rFoxM1-p1 | 294 (245–354) | 920 (480–1940) |
| r6T | 114 (103–127) | 704 (323–1668) |
| **DNA** | | |
| d6T | 370 (343–401) | 2097 (1494 – >2000) |
| d6T* | 509 (470–553) | >2000 |

RRACH at 3'-end), c-Myc-p5 (8-nt), and c-Myc-p6 (11-nt, RRACH at 5'-end), were derived from the *FOXM1* and *MYC* genes because of their involvement in m⁶A-mediated processes in glioblastoma and acute myeloid leukemia (*Vu et al., 2017*; *Zhang et al., 2017*). We covalently attached a fluorescein moiety at the 5'-end of all these oligos to facilitate the quantitative measurement of their binding affinities (equilibrium dissociation constant or Kd) to METTL3-METTL14 (see Materials and methods section for details).

We used a radiometric assay and fluorescence polarization (FP) to determine methylation activity and RNA binding, respectively. As shown in *Figure 2a and c*, RNA oligos with high propensity toward forming secondary structures (rNEAT2, rTCE23, Pal-Top+ bot) showed low to negligible levels of methylation by METTL3-METTL14. The other RNA oligos showed some levels of methylation, though they could not be methylated with as much efficiency as the ssDNA substrate, d6T*. The results of binding experiments are shown in *Figure 2b* and *Table 1*. The full-length METTL3-METTL14 binds to rNEAT2 with the highest affinity (Kd = 2 nM), followed by the hairpin rTCE23 (Kd = 25 nM), the 3'-over-hang r6T (Kd = 114 nM), the palindromic Pal-Top+ bot (Kd = 127 nM), and the ssRNA FoxM1-p1 (Kd = 294 nM). Surprisingly, the DNA substrates (d6T and d6T*) that showed highest methyltransferase activity were the poorest binders, with Kd values of 370 and 509 nM, respectively.

If METTL3-METTL14 were to methylate ssDNA in vivo, and the evidence to date does not support its direct enzymatic origin (*Musheev et al., 2020*), then what could be the significance of high affinity RNA binding to structured RNA oligos (rNEAT2 and rTCE23)? One explanation is that structural motifs in ncRNA or mRNAs can serve as a recruitment platform for METTL3-METTL14 in a specific context. But how would the RNA-binding activity then affect DNA methylation activity? To answer this question, we measured the methyltransferase activity of METTL3-METTL14 in the presence of roughly equimolar mixtures of d6T* DNA with each RNA oligonucleotide. The methyltransferase activity of METTL3-METTL14 was significantly reduced in the presence of RNA (*Figure 2c*). Of note, RNAs with propensity to form secondary structures and highest binding affinity to METTL3-METTL14 (rNEAT2, rTCE23) attenuated methyltransferase activity to the most negligible levels (*Figure 2c*). To further explore the relationship between methyltransferase activity and substrate-binding affinity, we synthesized a variant of the stem-loop oligo (rNEAT2*) with the DRACH motif (5'-AAACA-3') changed to an ideal m⁶A target sequence, that is, 5'-GGACU-3' (*Figure 1g*). Although higher methylation levels were observed for rNEAT2* compared to rNEAT2, most likely attributed to a perfect GGACU motif in rNEAT2*, yet the methylation was significantly lower than that of d6T* DNA (last panel of *Figure 2c*). Moreover, rNEAT2* showed ~6 fold reduction in affinity compared to rNEAT2 (*Figure 2b*). Importantly, rNEAT2* also attenuated methylation of the single-stranded d6T* DNA by METTL3-METL14 but to a lesser extent than the rNEAT2 (*Figure 2c*). These results suggests that: (a) the sequence and shape of the target RNAs dictate the m⁶A activity of METTL3-METTL14, and (b) that higher affinity substrates tend to show lower methylation levels, possibly due to slow off rates of m⁶A RNA.

To further investigate RNA-mediated inhibition, we showed that the two structured RNAs (rTCE23 and rNEAT2) inhibited the DNA methylation activity of METTL3-METTL14 in a dose-dependent manner, with rNEAT2 ($IC_{50}$ = 216 nM) showing 4-fold stronger inhibition than rTCE23 ($IC_{50}$ = 887 nM) (*Figure 2e*). Next, we employed an LC/MS-based oligonucleotide intact mass analysis to further validate the results of the radiometric assay. As shown in *Figure 2f*, an equimolar (with respect to d6T* DNA) addition of rNEAT2 or rTCE23 significantly reduced the methyltransferase activity from 80.5–14.5% to 7.3%, respectively. We then performed nucleoside analysis of the METTL3-METTL14 reactions to unambiguously determine the identity of methylated base and efficiency of methylation (*Figure 2g*). We confirmed that deoxyadenosine on d6T* and adenosine on rNEAT2 were the only modified bases. In the absence of rNEAT2 or rTCE23, 86.3% of dAs on d6T* were modified. When rNEAT2 or rTCE23 were present, only 14.4% and 7.4%, respectively, of dAs on d6T* were modified. Additionally, 5.8% of the rAs on rNEAT2 were modified. As expected, no methylation on rTCE23 was detected (no RRACH motif). These results are consistent with those obtained by oligonucleotide intact mass analysis. Altogether, the methyltransferase and binding assays confirm that the binding to structured RNAs, especially those lacking the GGACU sequence, almost completely abolishes the methyltransferase activity METTL3-METTL14.

The C-terminal RGG repeat motif of METTL14 contributes to RNA binding and activity of METTL3-METTL14 (*Schöller et al., 2018*; *Figure 3a*). Thus, we hypothesized that the absence of RGG motif in METTL3-METTL14 should diminish its ability to bind RNA and consequently attenuate the effect

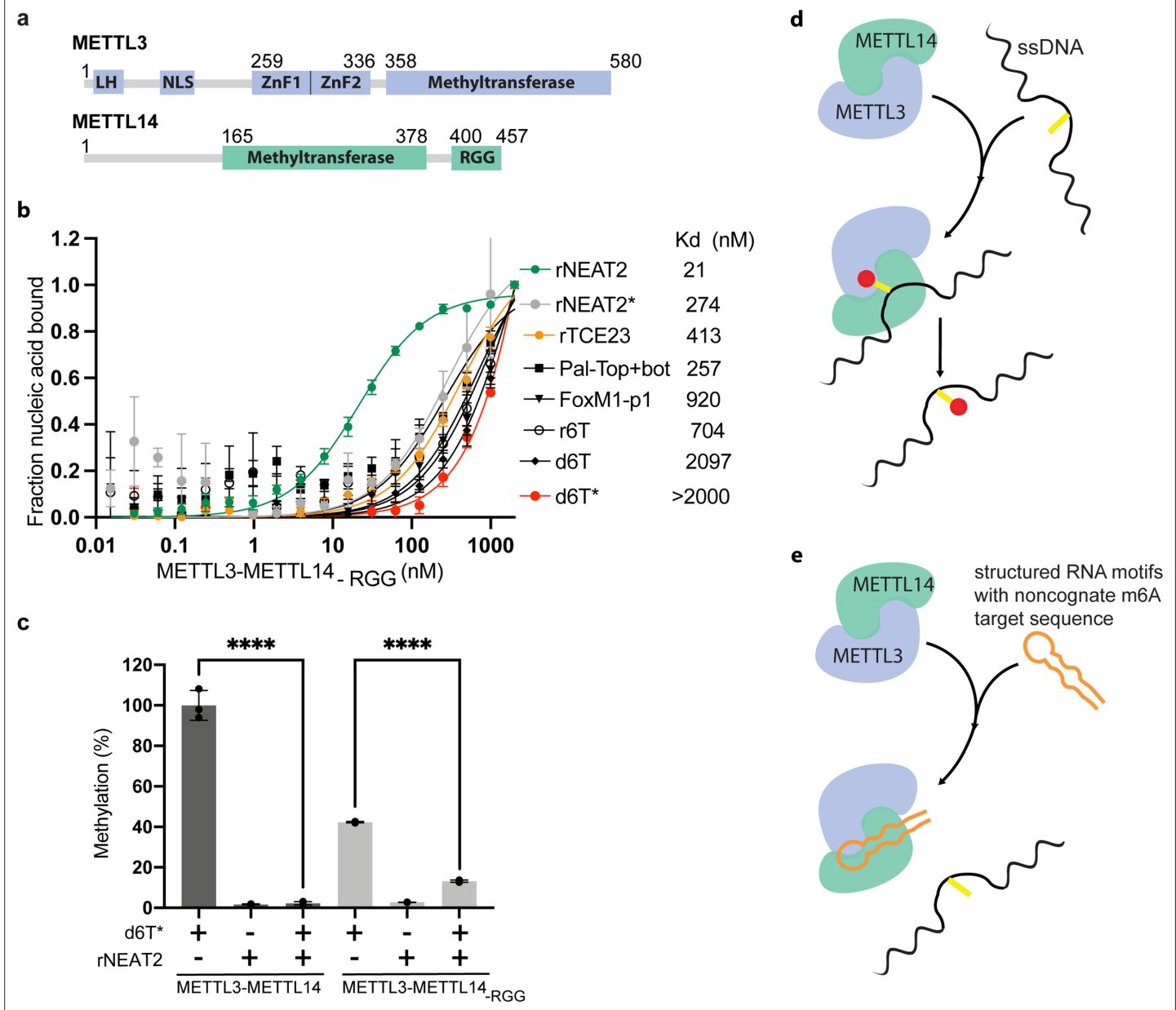

**Figure 3.** Role of RGG motifs and model of RNA-mediated regulation of methyltransferase activity. (**a**) Domain architecture of methyltransferase like-3 (METTL3) and METTL14. LH, leader helix; NLS, nuclear localization signal; ZnF1/2, zinc-finger domain 1/2; RGG, arginine-glycine rich repeats motif. (**b**) FP-based-binding assay for DNA and RNA oligos showing the highest affinity of METTL3-METTL14$_{(-RGG)}$ for rNEAT2 RNA (green) and lowest affinity for d6T* (red). Equilibrium dissociation constants (Kd) for each oligo are shown. The data were fit into one site-specific binding model (Y = Bmax*X/ (Kd+ X)). See Materials and methods section and source data for details. (**c**) Relative methyltransferase activity of full-length METTL3-METTL14 and the truncated enzyme devoid of the RGG motif in METTL14 ([METTL3-METTL14$_{(-RGG)}$]) in the presence of d6T*, rNEAT2, or an equimolar mixture of these two oligos, as measured by radiometric assay. Results presented are the average of three independent experiments (n = 3) with one standard deviation (s.d.) for each oligonucleotide (shown as error bars). The results of two groups were analyzed and compared using two-tailed Student's unpaired t-test (p-value < 0.0001). Details about Student's t-test are provided in the Source Data file. (**d, e**) Proposed models showing that the METTL3-METTL14 complex can methylate a target adenine (yellow) in a single-stranded DNA region (black). Structured motifs present in ncRNA/mRNAs (orange) can block the methyltransferase activity by a shape-dependent binding of these RNAs to METTL3-METTL14.

The online version of this article includes the following source data for figure 3:

**Source data 1.** Binding isotherms of METTL3-METTL14 (-RGG).

**Source data 2.** Relative MTase activity of METTL3-METTL14 and METL3-METL14 (-RGG).

of RNA on DNA methylation. In fact, we observed a 2- to 10-fold decrease in binding affinity by METTL3-METTL14$_{-RGG}$ (*Figure 3b* and *Table 1*). This suggests that the RGG motifs play a major role in RNA and DNA substrates, while other parts of the enzyme (e.g., CCCH-type zinc finger domains in METTL3 *Huang et al., 2019a* and MTase core of METTL14 *Wang et al., 2016a*) may also contribute to the overall binding (*Figure 3b*). As expected, the deletion of RGG caused 60% reduction in DNA methylation activity. In the presence of rNEAT2, the activity of METTL3-METTL14$_{-RGG}$ on d6T* DNA was reduced to about 75% of its activity in absence of the RNA; comparatively, a >90% activity reduction was observed for the full-length METTL3-METTL14 in presence of the same RNA (*Figure 3c*). These results suggest that the RGG motif, as a general RNA-binding domain, dictates the shape-specific RNA recognition and promotes an RNA-mediated restriction of the DNA methylation activity of METTL3-METTL14 (*Table 1*).

## Discussion

Our results are noteworthy given the relevance of METTL3-METTL14 in regions where RNA and DNA are held in closed proximity, as supported by recent studies suggesting: (a) localization of METTL3 to sites of dsDNA breaks, and its role in DNA/RNA hybrid accumulation (*Zhang et al., 2020*), (b) recruitment of METTL3 to chromatin by promoter-bound transcription factors (*Barbieri et al., 2017*), (c) UV-induced DNA damage signaling (*Xiang et al., 2017*), and (d) metabolism of R-loops (*Abakir et al., 2020*). Earlier studies suggest the existence of a large complex of multiple subunits of ~1 MDa size, possessing RNA $N^6$-adenosine methylation activity distributed over two sub-complexes of approximate molecular mass of ~200 and ~800 kDa (*Bokar et al., 1994Bokar et al., 1997*). One such subunit is WTAP, which alters both the pattern and location of the m$^6$A transcriptome (*Schwartz et al., 2014*). Future studies could examine the role of WTAP and other partners of METTL3-METTL14 in substrate recognition and specificity, especially within the context of the regulatory role of RNA in DNA methylation.

In summary, we unveiled a new $N^6$-deoxyadenosine methylation activity on ssDNA, which is regulated by binding of METTL3-METTL14 to shape-dependent structured regions of ncRNAs and chromatin-associated nascent pre-mRNAs (CA-RNA) (*Figure 3d and e*). The structured elements in RNA could facilitate the recruitment of METTL3-METTL14 to chromatin and/or keep the writer complex engaged during co-transcriptional occurrence of m$^6$A. Such a topological arrangement of METTL3-METTL14/RNA interaction could also restrict m$^6$dA deposition, especially to the single-stranded regions of DNA exposed during transcription (*Slobodin et al., 2017*), DNA recombination (*Zhang et al., 2020*), damage and repair (*Xiang et al., 2017*), and R-loop metabolism (*Abakir et al., 2020*).

A major difference in prokaryotic and eukaryotic genomes is the contrasting occurrence of the base methylation. The m$^6$dA is a predominant modification mark in bacterial DNA, whereas the m$^5$dC is more prevalent in eukaryotic genomes. How is the eukaryotic genome then protected from m$^6$dA deposition despite having an efficient enzyme machinery (e.g., METTL3-METTL14) to accomplish such task? A comparative sequence/structural analysis of the bacterial EcoP15I and human METTL3 and METTL14 MTases provides some hints (*Figures 1a–c and 3a*). Of note, there are two major differences that exist in the sequences of METTL3 and METTL14: (a) loss of the target recognition domain (TRD) and (b) acquisition of the RGG motif in METTL14. The TRD motif (in EcoP15I MTase) is a major contributor to binding and specificity for a dsDNA (*Gupta et al., 2015*), whereas the RGG motif is known for RNA binding (*Thandapani et al., 2013*). It is likely that gene fusion/genome arrangement event(s) could have occurred during evolution (*Bujnicki et al., 2002*; *Woodcock et al., 2020*) to ensure very low/no occurrence (*Douvlataniotis et al., 2020*) of m$^6$dA in mammalian genome. The high affinity to structured RNAs as demonstrated here could also ensure efficient recruitment of METTL3-METTL14 to specific nuclear compartment(s) in human cells.

Interestingly, a recent study showed that a comprehensive network of RNA-RNA and RNA-DNA interactions mediated by ncRNAs (including MALAT1) creates distinct nuclear compartments and facilitates recruitment of proteins for RNA processing, heterochromatin assembly, and gene regulation (*Quinodoz et al., 2020*). Thus, the secondary structure and 3D shape of the RNAs appears to be crucial for these important biological processes. In line with these observations, our results suggest that structured elements in RNAs may play an important role in regulating the ssDNA and ssRNA methylation activity of METTL3-METTL14.

# Materials and methods

## Protein expression, purification, and oligonucleotide preparation

The coding sequence of full-length human METTL3 (NCBI reference sequence GI: 33301371) was used in this study. The gene was cloned into a plasmid suitable for expression of proteins in insect cells with an N-terminal poly histidine tag followed by a tobacco etch virus (TEV) protease site. This plasmid (5TEY (METTL3)) was a kind gift from Dr Cheryl Arrowsmith (Addgene plasmid # 101892; http://n2t. net/addgene:101892; RRID:Addgene_101892). The coding sequence of full-length human METTL14 (NCBI reference number GI: 172045930) was cloned into the pFastBac1 vector between BamHI and NotI restriction enzyme sites. For expression of recombinant proteins, we used ExpiSf Expression System (Thermo Fisher). First, the viral DNA bacmids for each gene were prepared from individual plasmids transformed into MAX Efficiency DH10Bac competent cells (Thermo Fisher). We confirmed the identity of the inserted genes by PCR amplification and DNA sequencing. The recombinant P0 virus for each gene were generated in ExpiSf9 insect cells following the manufacturer's recommendations (Thermo Fisher). The amount of each virus needed for infection has been optimized for maximal production of the two proteins. Infected cells were grown in ExpiSf CD medium (Thermo Fisher) at 27°C on an orbital shaker (speed 125 rpm) in a controlled (non-humidified, air-regulated) environment. Cells were harvested at 72 hr post-infection by spinning at 300× g for 5 min, washed with phosphate buffered saline (PBS) twice, and resuspended in a buffer A (0.025 M Tris-HCl pH 8.0, 0.5 M NaCl, 0.005 M imidazole, 0.1 mM TCEP, 10% [(v/v)] glycerol) supplemented with 0.5% (v/v) Igepal, two mini-protease inhibitor cocktail tablets (Roche), DNase I, and lysozyme. Cells were lysed using a microfluid-izer (Analytik, UK) and the soluble fraction was separated by centrifuging the lysate at 40,000 rpm for 40 min. The supernatant was passed through a 0.22 μm filter.

The clarified soluble fraction was loaded on to a Nuvia IMAC column (Bio-Rad) pre-equilibrated in buffer A. The proteins were eluted by increasing the concentration of imidazole. The 6XHis tag was then proteolytically removed using TEV protease. Any uncleaved fractions and remaining protein impurities were removed by second passage through an IMAC column. The METTL3-METTL14 complex was further purified by successive passage through HiTrap heparin and HiLoad Superdex 200 columns (GE Healthcare). The protein complex was eluted as a single homogenous species in a final buffer containing 0.02 M Tris-HCl pH 8.0, 0.15 M NaCl, 0.1 mM TCEP. The purified protein complex was concentrated to 3–5 mg/mL and used immediately, or flash-frozen in liquid nitrogen and stored at –80°C. The METTL3-METT14$_{-RGG}$ complex lacks the C-terminus RGG repeats motif (amino acids 400–457) of METTL14. It was expressed and purified using same method as the full-length METTL3-METTL14 complex.

All RNA and DNA oligonucleotides used in this study were synthesized, purified by HPLC, and received in deprotected (for RNAs) and desalted form from Dharmacon and Integrated DNA Technologies, respectively. Oligos were dissolved in 1× buffer containing 0.01 M Tris-HCl pH 8.0, 0.05 M NaCl. To prepare double-stranded oligos, the two strands were mixed in roughly equimolar amounts. Annealing was carried out by heating the mixture to 95°C for 2 min followed by gradual cooling to 25°C over 45 min in a thermocycler (Eppendorf).

## Methyltransferase assays

### Radiometric assay

Each reaction was carried out in a 5 μL mixture containing 50 mM HEPES pH 7.5, 5 mM NaCl, 1 mM dithiothreitol (DTT), 5 μM [methyl-$^3$H] SAM (PerkinElmer), 10 μM substrate RNA/DNA probe, and 1 μM co-purified full-length METTL3-METTL14 enzyme complex. The reactions were incubated at 37°C for 1 hr and 3 μL of each reaction was quenched by blotting on Hybond-N+ membrane (Amersham). RNA/DNA probes were crosslinked to membrane by exposure to ultraviolet light (254 nm) for 2 min. The membranes were successively washed three times by 1× PBS followed by three ethanol washes for 5 min each. The membranes were air-dried in the hood for 15 min and the counts per minute (c.p.m.) of the RNA/DNA probes were measured using a scintillation counter (Beckman LS6500). We also performed this assay in a reaction buffer containing higher salt (50.0 mM NaCl) and observed consistent results (dose-dependent inhibition of m$^6$dA activity of METTL3-METTL14 by small, structured RNA oligos) as presented in *Figure 2e*. Results presented here are an average of three independent experiments (n = 3) with one standard deviation (s.d.) for each oligonucleotide shown as error bars.

Source data are provided as a Source Data file. The results of two groups in *Figure 3c* were derived from three independent experiments and were analyzed using two-tailed Student's t-test (unpaired).

## Quantitation of methylation and validation of identity of methylated base by mass spectrometry

Methylation reactions were carried out in a 10 µL mixture containing 50 mM Tris-HCl, pH 7.5, 5 mM NaCl, 1 mM DTT, 200 µM SAM, 10 µM substrate RNA/DNA probe, and 1 µM METTL3-METTL14 enzyme complex. The reactions were incubated at 37°C for 1 hr, followed by incubation with 0.8 units of proteinase K (New England Biolabs) at 37°C for 10 min. Two microliters of each reaction were subjected to oligonucleotide intact mass analysis by liquid chromatography-mass spectrometry (LC-MS) as described previously (*Viswanathan et al., 2020*). The raw data was deconvoluted using ProMass HR (Novatia, LCC). The deconvoluted-mass peak area ratio between reactants and expected products was used to estimate the percentage of methylation. Results from three independent experiments (n = 3) are shown. To verify the identity of the modified nucleotides, METTL3-METTL14 reactions were also subjected to nucleoside analysis. Briefly, 8 µL of each of reaction were purified using Oligo Clean-Up and Concentration Kit (Norgen Biotek) according to manufacturer's instructions. The nucleic acids were eluted in 20 µL nuclease-free water. To seventeen microliters of the eluates were added 2 µL of 10× Nucleoside Digestion Mix reaction buffer and 1 µL of Nucleoside Digestion Mix (New England Biolabs). Reactions were incubated at 37°C for 1 hr. The entirety of the nucleoside digestion reaction was subjected to ultra-high-performance liquid chromatography (UHPLC) or UHPLC-MS analysis without purification. UHPLC analysis was performed on an Agilent 1290 Infinity II system equipped with a G7117A diode array detector (DAD). UHPLC-MS analysis was performed an Agilent 1290 Infinity II system equipped with a G7117A DAD and a 6135 XT mass detector. LC separations were carried out using a Waters XSelect HSS T3 XP column (2.1 × 100 mm, 2.5 µm) with the gradient mobile phase consisting of methanol and 10 mM ammonium acetate buffer (pH 4.5). The identity of each nucleoside was confirmed by analysis of the LC retention times relative to authentic standards and by mass spectrometric analysis. The relative abundance of each nucleoside was determined by UV absorbance.

## Determining the nucleic acid-binding affinities of METTL3-METTL14 and METTL3-METTL14$_{-RGG}$

FP-based assay was used to measure equilibrium dissociation-binding constants (Kd). The reactions were carried out in a buffer 0.01 M HEPES pH 7.5 and 0.05 M KCl. All oligonucleotides contain a fluorescein moiety covalently attached to the 5'-end. A constant concentration of the fluorescein labeled oligo (1 nM) was used with increasing concentrations of full-length METTL3-METTL14 protein or its truncated version (METTL3-METTL14$_{-RGG}$ (0–2000 nM)) in a 384-well plate. Significant changes in FP upon increasing concentrations of protein were indicative of direct binding. The FP (emission wavelength = 530 nm, excitation wavelength = 485 nm) value for each dilution was measured using PHERAstar FS (BMG Labtech). The buffer corrected values were used to calculate the equilibrium dissociation constant (Kd) for protein-DNA/RNA binding using a simple 1:1 specific binding model ($Y = Bmax*X/(Kd+ X)$, where X denotes the concentration of protein ligand, Y = specific binding, Bmax = maximum binding, Kd = equilibrium dissociation constant in same units as X). Data were fitted by a single-site binding model in GraphPad Prism (GraphPad Software, San Diego, CA). Results presented here are an average of three independent experiments (n = 3) with one standard deviation (s.d.) for each oligonucleotide shown as error bars. Source data are provided as a Source Data file.

## Acknowledgements

This work was partly supported by funding from the Max and Minnie Tomerlin Voelcker Foundation, San Antonio Partnership for Precision Therapeutics, IIMS/CTSA pilot award, and laboratory startup funds from the Greehey Children's Cancer Research Institute (GCCRI) of UT Health San Antonio to YKG. SQ is supported by the GCCRI. YKG is a recipient of a high impact/high risk award (RP190534) from the Cancer Prevention and Research Institute of Texas (CPRIT), and a Rising STARs award from the University of Texas System. YKG and MKR are supported by CPRIT grant

(RP200110). YKG is grateful for the support from NIH grant 1R01AI161363-01. S-HC, ND, and IRC are grateful to Donald Comb, Jim Ellard, and Rich Roberts for their support of research at New England Biolabs.

## Additional information

### Competing interests

Siu-Hong Chan, Nan Dai, Ivan R Corrêa Jr: is an employee of New England Biolabs, a manufacturer and vendor of molecular biology reagents. Robert A Hromas: owns equity in Dialectic Therapeutics and Abfero. Yogesh K Gupta: is founder of Atomic Therapeutics. None of these affiliations affect the authors' impartiality, adherence to journal standards and policies, or availability of data. The other authors declare that no competing interests exist.

### Funding

| Funder | Grant reference number | Author |
| --- | --- | --- |
| Max and Minnie Tomerlin Voelcker Fund | | Yogesh K Gupta |
| IIMS/CTSA pilot award | UL1 TR002645 | Yogesh K Gupta |
| Greehey Children's Cancer Research Institute | | Shan Qi Yogesh K Gupta |
| National Institute of Allergy and Infectious Diseases | 1R01AI161363-01 | Yogesh K Gupta |
| Cancer Prevention & Research Institute of Texas | RP190534 | Yogesh K Gupta |
| CPRIT | RP200110 | Yogesh K Gupta Manjeet K Rao |
| NIH | 1R01AI161363 | Yogesh K Gupta |
| New England Biolabs | | Siu-Hong Chan Nan Dai Ivan R Corrêa |
| University of Texas System | Rising STARs award | Yogesh K Gupta |

The funders had no role in study design, data collection and interpretation, or the decision to submit the work for publication.

### Author contributions

Shan Qi, Data curation, Formal analysis, Investigation, Methodology, Validation, Visualization; Javier Mota, Data curation, Formal analysis, Investigation, Methodology, Visualization; Siu-Hong Chan, Ivan R Corrêa Jr, Data curation, Formal analysis, Investigation, Methodology, Writing – review and editing; Johanna Villarreal, Shailee Arya, Methodology; Nan Dai, Formal analysis, Investigation, Methodology; Robert A Hromas, Manjeet K Rao, Resources; Yogesh K Gupta, Conceptualization, Data curation, Formal analysis, Funding acquisition, Investigation, Methodology, Project administration, Resources, Supervision, Validation, Visualization, Writing – original draft, Writing – review and editing

### Author ORCIDs

Shan Qi http://orcid.org/0000-0003-0175-6267
Ivan R Corrêa Jr http://orcid.org/0000-0002-3169-6878
Yogesh K Gupta http://orcid.org/0000-0001-6372-5007

### Decision letter and Author response

Decision letter https://doi.org/10.7554/eLife.67150.sa1
Author response https://doi.org/10.7554/eLife.67150.sa2

## Additional files

### Supplementary files
• Transparent reporting form

### Data availability
The information about coding sequences of human METTL3 (NCBI reference sequence GI: 33301371) and METTL14 (NCBI reference sequence GI: 172045930) used in this study is available at NCBI. Source data are provided as a separate Source Data file. Correspondence and requests for material should be addressed to Y.K.G. (guptay@uthscsa.edu).

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
