## [Editor Report]

This manuscript will be of interest to researchers in the fields of nucleic acid chemical biology in general and diseases related to nucleic acid methylation in particular. The data presented support the conclusions of the paper within the current context, provide new evidence and plausible explanations to previously inexplicable mechanisms.

---

## [Decision Letter]

**Decision letter after peer review:**

Thank you for resubmitting your work entitled "RNA binding to human METTL3-METTL14 restricts N6-deoxyadenosine methylation of DNA" for further consideration by *eLife*. Your article has been reviewed by 3 peer reviewers, one of whom is a member of our Board of Reviewing Editors, and the evaluation has been overseen by Mone Zaidi as the Senior Editor. The reviewers have opted to remain anonymous.

Essential revisions:

Please address a few questions raised by the Reviewer #1 #2 and #3. Their comments are appended.

*Reviewer #1:*

The manuscript submitted by Shan Qi et al., investigates the mechanistic role of METTL3-METTL14 enzymes in the methylation of DNA and RNA. Up until now, the detailed understanding of their specificity for nucleic acids has been liimted. Their study uses classical biochemical and mass spectrometry experiments to investigate the relationship between substrate binding and methylation. Interestingly, their results highlight that the METTL3-METTL14 complex prefers to methylate dA in single-stranded DNA even though it does not prefer to bind to DNA. They also go on to demonstrate that the complex binds structured RNA with high affinity but methylates RNA less efficiently than single-stranded DNA and that structured RNA prevalent in long noncoding RNA regulates methylation of single-stranded DNA.

Strength:

This work adds to the discussion about the substrate specificity of these enzyme complexes. The data presented support the conclusions made.

Weakness:

Since the authors use structural models, they should detail the structural aspects of the METTL3-METTL14 complex behind the preference of one substrate over the other.

The reviewer believes that the study will benefit immensely if the structural details are expanded to explain substrate specificity in their discussion. The authors use structural models in their figures but do not use these models to their advantage in explaining their results. This should be included.

*Reviewer #2:*

In this manuscript by Qi et al., the authors have studied the methyltransferase activity of the METTL3-METTL14 complex. The authors propose that the structured elements in RNAs may play an essential role in regulating the ssDNA and ssRNA methylation activity by METTL3-METTL14 and affect biological processes where DNA and RNA are held in close proximity such as transcription, DNA recombination, transcription-coupled DNA damage repair, and R-loop metabolism.

To carry out this study, the authors have purified full-length METTL3-METTL14 and METTL3-METTL14 truncated protein lacking RGG motif and measured its methyltransferase activity and binding affinity with different DNA and RNA substrates. Differential binding affinity was observed for different DNA and RNA oligos which were fluorescently labeled for the binding assay.

The results demonstrate an inverse relationship for substrate binding and methyltransferase activity in ssDNA and RNA in 'in vitro' experiments. While the M3-M14 complex showed the highest affinity and lowest activity for structured RNA oligos, maximum activity and lowest affinity were observed with single-stranded DNA. And the presence of structured RNA oligos can modulate the methyltransferase activity of the M3-M14 enzyme on single-stranded DNA.

Before this research, it was reported that the METTL3-METTL14 complex from mammals could catalyze the transfer of methyl group to N6 of Adenine in ssDNA and unpaired DNA in vitro but spares the double-stranded DNA or DNA/RNA hybrid (Woodcock et al., 2019). Further, it was also shown that the catalytic efficiency on ssDNA is much higher than that on RNA substrate. Moreover, Wilms tumor suppressor-associated protein (WTAP) interacts with this complex and affects its methylation activity.

The enzymatic reactions performed in this work are at non-physiological conditions with extremely low salt. The authors have discussed the significance of the work in transcription, DNA damage repair, and R loop metabolism. However, the substrates used in this study do not represent the process mentioned. Hence, it is difficult to reach rigorous insights into the biological processes for the activity observed.

The subject of the work is interesting; however, the progress made to the field is incremental. There some concerns which I have listed below:

1. The enzymatic reactions performed in this work are at non-physiological conditions in 5mM NaCl. Hence, it is difficult to comment on the biological processes for the activity observed under such conditions. The title is misleading and it should state 'in vitro'. In the abstract, the authors state: "We also show that the methyltransferase activity of METTL3-METTL14 on ssDNAs is largely regulated by structured RNA elements prevalent in long noncoding (lnc) RNAs but also found in other cellular RNAs." However, the experiments are done in vitro using short oligos. Also, WTAN is known to interact with this complex and affect its methylation activity. In the absence of an experiment to test the role of WTAN in the methyltransferase activity of METTL3-METTL14, it may be difficult to reach this conclusion. The authors have discussed the significance of the work in the process of transcription, DNA damage repair, and R loop metabolism, and also depicted in the model in Figure 3d. However, none of the substrates used in this study represent that of the model.

2. The enzymatic reactions are done in buffer with 5 mM NaCl, while binding studies were done in 50 mM KCl. Why the difference in salt concentrations? What role the different salt conditions play in these experiments?

3. Why rNEAT2 binds with higher affinity but has lower methylation than r6T, which binds with lesser affinity? What is the mechanism behind the inverse relationship of binding and methylation activity?

4. The authors suggest that 'the code for shape-specific RNA recognition resides in the RGG motif'. So in Figure 3c, why the % methylation in d6T* decreases significantly in RGG mutant compared to wild type. Why is an affinity change observed for unstructured d6T* and r6T in RGG mutant in Table 1?

5. In the present form, the manuscript is difficult to read for a general reader. The manuscript text assumes the reader has a more detailed understanding of the subject. For example, the RGG motif and its role in substrate binding is not introduced; the structural similarity between human METTL3-METTL14 and N6-deoxyadenosine DNA methyltransferases.

*Reviewer #3:*

Qi et al., have investigated the RNA/DNA substrate specificity of human Methyltransferase like-3 (METTL3) and METTL14 complex using various substrates including ssDNA, dsDNA, various RNA substrates with or without secondary substrate and various placements of the specific recognition motif DRACH. They showed higher methyltransferase activity towards ssDNA (d6T*) compared to a previously identified DNA substrate d6T which had the propensity to form duplex DNA or its RNA counterpart. They also show an inverse relation between binding affinity and catalysis with higher affinity to RNA substrates with propensity to form secondary structures with negligible levels of methylation. Similarly, the DNA substrates with the 370 and 509nM Kd for d6T and d6T* had the highest methyltransferase. They further investigate the effect of structured RNA binding on DNA methylation, showing a inhibitory effect in a dose-dependent manner (a 4-fold stronger inhibition with rNEAT2) compared to a control RNA substrate with no recognition motif. They further investigate the effect of RGG deletion (a previously identified RNA binding region) which reduces RNA substrate binding by 2- to 10-fold with 75% reduction in DNA methylation activity compared to >90%. The authors suggest the role of structured RNAs for recruitment of the METTL3/METTL14 complex to specific sites for function. This manuscript provides incremental advances in further qualifying the substrate specificity under the experimental conditions tested.

The main conclusions of the paper on the role of structured RNAs for DNA methylation are only partially supported in this paper in the context of the experiments carried out. A wider mechanistic understanding of the role of structured RNAs for DNA methylation or why a ssDNA substrate is preferable over a RNA substrate when the occurrence of DNA methylation is a lot fewer than RNA methylation are all aspects that cannot be addressed by the specific experiments in this paper and is a particular weakness in this manuscript.

To enhance the manuscript and further the conclusions in this manuscript, authors need to further investigate the role of structured RNAs initially using molecular modelling to further inform functional roles. The authors in Figure 1 show structure of EcoP15I with a DNA substrate to show a model of the METTL3/METTL14 with an RNA substrate. It is not clear how the authors have gone about modelling this, what is the sequence similarity/ conservation of residues in the binding site, is the RGG domain involved in binding in this model, does the structural model support specificity to RNA or DNA, which residues might be involved in the active site? Authors need to undertake a structural analysis using the crystal structure available to perform molecular dynamics with DNA and RNA substrates to get a mechanistic understanding for this or do an experimental structure with a nucleic acid substrate to provide this understanding.

The authors have tested the effect of structured RNAs on DNA methylation specifically, why not RNA substrates lacking structure. Is this a competition between substrates with different affinities and what could be the relevance in cellular processes specifically? Competition experiments haven't been performed here.

---

## [Author Response]

Reviewer #1:[…]The reviewer believes that the study will benefit immensely if the structural details are expanded to explain substrate specificity in their discussion. The authors use structural models in their figures but do not use these models to their advantage in explaining their results. This should be included.

As shown in the revised Figures 1a-e and supplementary figure. 1, we have made a detailed comparison of the methyltransferase domains of EcoP15I, METTL3, and METTL14. We have added a description of these models to explain our results on pages 3 (lines 50-54) and 5 (lines 89-97).

Reviewer #2:[…]1. The enzymatic reactions performed in this work are at non-physiological conditions in 5mM NaCl. Hence, it is difficult to comment on the biological processes for the activity observed under such conditions.

We followed the same protocol in a previous study that reported, for the first time, the m6dA methyltransferase activity of METTL3-METTL14 (Woodcock et al., Cell Discovery 2019; PMID: 31885874). In this study they used a 14-mer DNA substrate (d6T). Since we wanted to measure the methyltransferase activity of METTL3-METTL14 on different RNA/DNA substrates and compare it to its activity on the original DNA substrate (d6T) used in the study by Woodcock et al., we used their identical experimental conditions to minimize experimental bias. Moreover, an independent research laboratory tested the enzymatic activity of METTL3-METTL14 in different concentrations of NaCl and found highest activity at low NaCl (0 – 5mM NaCl) concentration. Please see Figure 1C from Li et al., J of Biomol Screen. 2016 Mar; 21(3):290-7. PMID:26701100 (https://journals.sagepub.com/doi/10.1177/1087057115623264?url_ver=Z39.88-2003&rfr_id=ori%3Arid%3Acrossref.org&rfr_dat=cr_pub++0pubmed&).

Regardless, we performed the inhibition of m6dA activity of METTL3-METTL14 by both rTCE23 and rNEAT2 RNA in a buffer that contains 50.0 mM NaCl. As shown in panel B of Figure 2e, both RNA oligos inhibited the m6dA activity, with only modest increment in IC_50_ values at a higher salt concentration (50.0 mM NaCl). These results further strengthen our conclusion about ability of small structured RNAs to block m6dA activity in vitro*,* which is consistent at higher salt concentration. We have added the new results (panel B) in the lower panel of the revised Figure 2e.

The title is misleading and it should state 'in vitro'.

We agree with the reviewer and added *‘*in vitro*’* to the title as suggested.

In the abstract, the authors state: "We also show that the methyltransferase activity of METTL3-METTL14 on ssDNAs is largely regulated by structured RNA elements prevalent in long noncoding (lnc) RNAs but also found in other cellular RNAs." However, the experiments are done in vitro using short oligos. Also, WTAN is known to interact with this complex and affect its methylation activity. In the absence of an experiment to test the role of WTAN in the methyltransferase activity of METTL3-METTL14, it may be difficult to reach this conclusion. The authors have discussed the significance of the work in the process of transcription, DNA damage repair, and R loop metabolism, and also depicted in the model in Figure 3d. However, none of the substrates used in this study represent that of the model.

METTL3-METTL14 enzymes have been studied extensively as RNA methyltransferases (m6A). Most of these major studies have delineated the physiological roles of this complex in human biology and diseases by genetic depletion of METTL3 and/or METTL14. METTL3-METTL14 can form up to ~1.0 mega Dalton size complex through its direct interaction with WTAP (as pointed out by the reviewer), and other accessory proteins (through WTAP) e.g., RBM15/15B, VIRMA, CBLL1 (HAKAI), ZC3H13 (Bokar et al., J Biol Chem. 1994 Jul 1;269(26):17697-704; PMID: 8021282), (Zaccara et al., Nat Rev Mol Cell Biol; 2019 Oct;20(10):608-624; PMID: 31520073). While WTAP is an important accessory factor of METTL3-METTL14, it also harbors an uncharacterized N-terminal domain followed by a largely disordered C-terminus. We could not test the effect of WTAP mainly due to technical difficulties associated with the purification of the tag-free full length human WTAP protein in large enough quantities and in stable form.

Woodcock et al., (Cell Discovery 2019; PMID: 31885874) who first reported the DNA methyltransferase (m6dA) activity of METTL3-METTL14 also did not use WTAP in their methyltransferase assay. Given the essential role played by METTL3-METTL14 in human embryonic development (Geula et al., Science. 2015 Feb 27;347(6225):1002-6; PMID: 25569111) it is important to study the factors that regulate its enzymatic activity. Thus, our major finding that structured RNA elements can block m6dA activity is very important and may provide strong rationale for new in vivo or genetic studies.

We agree with the reviewer that the substrates used in this work are not true representation of Rloops, and/or nucleic acids generated during DNA damage and repair. We have revised the figure 3d to rectify this concern. Moreover, we believe the addition of ‘in vitro’ to the revised title, as suggested by the reviewer, should alleviate the confusion, and convey our conclusions in a simplified form.

2. The enzymatic reactions are done in buffer with 5 mM NaCl, while binding studies were done in 50 mM KCl. Why the difference in salt concentrations? What role the different salt conditions play in these experiments?

A previous study has thoroughly examined the effects of various salts and concentrations and concluded that METTL3-METTL14 has comparable activities in both KCl and NaCl. Please see figure 1B and C from Li et al., J of Biomol Screen. 2016 Mar; 21(3):290-7. PMID: 26701100

(https://journals.sagepub.com/doi/10.1177/1087057115623264?url_ver=Z39.882003&rfr_id=ori%3Arid%3Acrossref.org&rfr_dat=cr_pub++0pubmed&)

We used NaCl in methyltransferase assay because we wanted to examine the inhibition of m6dA activity of METTL3-METTL14 by RNA oligos under identical assay condition that was previously used for assaying m6dA activity on a single stranded DNA (d6T) (Woodcock et al., Cell Discovery 2019; PMID: 31885874). Moreover, the same group has recently reported the METTL3-METTL14’s activity on double stranded RNA containing DNA lesions (Yu et al., Nucleic Acids Res. 2021 Nov 18;49(20):11629-11642; PMID: 34086966). In this study, they also used 5.0 mM NaCl in their methyltransferase reaction.

As mentioned in our response to previous comment, we did examine the m6dA activity of METTL3-METTL14 and its dose dependent inhibition by RNA oligos in a buffer containing higher salt (50.0 mM) and did not find major effect of higher salt concentration.

We routinely use KCl in the buffers for the binding studies performed by fluorescence polarization method. It is known that both sodium and potassium ions stabilize DNA with similar potency (Owczarzy et al., Biochemistry 2008 May 13;47(19):5336-53; PMID: 18422348). Thus, there was no specific reason for using KCl other than a routine practice in the lab.

3. Why rNEAT2 binds with higher affinity but has lower methylation than r6T, which binds with lesser affinity? What is the mechanism behind the inverse relationship of binding and methylation activity?

The rNEAT2 RNA oligonucleotide is predicted to form a stem-loop structure as shown in figure 1g whereas the r6T RNA can only form a partial duplex with 8 nucleotide long 3’- overhang, but no stem-loop. Binding of METTL3-METTL14 to structured RNA oligos that harbor imperfect (e.g., rNEAT2) or no m6A target sequence (e.g., rTCE23) is stronger than the unstructured nucleic acids while the r6T shows higher RNA methylation than rNEAT2. This inverse relationship of binding and methylation can be explained by differences in the off rate (K_off_) of METTL3-METTL14. It is likely that METTL3-METTL14 when complexed with rNEAT2 has slower off rate than r6T. In line with this observation, the d6T* DNA yielded highest methylation but weaker binding than both r6T and rNEAT2. In the revised manuscript we included additional data in Figure 2. We tested a mutated version of rNEAT2 termed as rNEAT2* where we replaced the DRACH motif (AAACA) in its loop region to an ideal m6A target sequence (GGACU) for METTL3-METTL13. The rNEAT2* showed higher RNA methylation than rNEAT2 but could not achieve the methylation levels of single stranded d6T* DNA (last panel of Figure 2c). Consistently, rNEAT2* showed 6-fold weaker binding (Kd = 13nM) to METTL3METTL14 than rNEAT2 (Kd = 2nM) (Figure 2b). These results are in line with our model that both shape and sequence of nucleic acids dictate the efficiency of SAM-dependent methylation of RNA and DNA by METTL3-METTL14. Consistently, the addition of rNEAT2* to the reaction mixture of METTL3-METTL14-d6T* attenuated the overall methylation thus, corroborate our conclusions in this study that structured RNAs restrict the m6dA methylation in ssDNA by METTL3-METTL14 in vitro. We have added this description on page 9-10.

4. The authors suggest that ‘the code for shape-specific RNA recognition resides in the RGG motif’. So in Figure 3c, why the % methylation in d6T* decreases significantly in RGG mutant compared to wild type. Why is an affinity change observed for unstructured d6T* and r6T in RGG mutant in Table 1?

As shown in Table 1 and Figure 3b, the binding of METTL3-METTL14 enzyme devoid of RGG motifs is uniformly reduced to both the DNA and RNA substrates. From these results we conclude that RGG motifs participate in substrate binding. We agree with the reviewer that our data do not reveal the code for shape-specific RNA recognition residing in the RGG motif. We have rephrased the sentence in the main text (on page 11 lines 228-231) to clarify this notion as follows:

“This suggests that the RGG motifs participate in the binding of both RNA or DNA, while other parts of the enzyme (e.g., CCCH-type zinc finger domains in METTL3 and METTL14 MTase) may also contribute to the overall binding of the substrate (Figure 3b).”

5. In the present form, the manuscript is difficult to read for a general reader. The manuscript text assumes the reader has a more detailed understanding of the subject. For example, the RGG motif and its role in substrate binding is not introduced

We agree with the reviewer and have rectified this by revising the main text. We have added a description of RGG motifs on page 5 (lines 102-109) as follows:

“The RGG motifs represent the clustered sequence of arginines and glycines. These motifs are commonly present in diverse set of RNA binding proteins that play important roles in physiological processes such as RNA synthesis and processing*^37^*. In human METTL14, a total of six tri-RGG motifs are two RG motifs are present at its c-terminus tail (aa 408-457). As expected, this region also harbors a few aromatic amino acids, which can further stabilize the nucleic acids via hydrophobic or stacking interactions. Consistently, this region contributes to the substrate binding and m^6^A activity of METTL3-METTL14 enzyme complex*^38^*. The sequence encompassing the RGG motifs in METTL14 is well conserved in higher vertebrates*^17^*.”

Reviewer #3:[…]To enhance the manuscript and further the conclusions in this manuscript, authors need to further investigate the role of structured RNAs initially using molecular modelling to further inform functional roles. The authors in Figure 1 show structure of EcoP15I with a DNA substrate to show a model of the METTL3/METTL14 with an RNA substrate. It is not clear how the authors have gone about modelling this, what is the sequence similarity/ conservation of residues in the binding site, is the RGG domain involved in binding in this model, does the structural model support specificity to RNA or DNA, which residues might be involved in the active site? Authors need to undertake a structural analysis using the crystal structure available to perform molecular dynamics with DNA and RNA substrates to get a mechanistic understanding for this or do an experimental structure with a nucleic acid substrate to provide this understanding.

As explained in our previous responses to the comments #4 (from Reviewer 2) and comment #1 (reviewer 1), we have included detailed description of our structural modeling and analyses. The substrate binding regions of EcoP15I and METTL-METTL14 are very different. For example, in EcoP15I a well folded target recognition domain (TRD, aa 263-384) is present in addition to the canonical MTase motifs (IV-X, and I-III) in the class β MTases. While METTL3 and METTL14 both have conserved canonical MTase motifs they both lack an extensive TRD as in EcoP15I. In METTL3-METTL14 the substrate recognition can be accomplished by the combination of three relatively short loops encompassing the sequence intervening MTase motifs IV and V (loop 1), VIII and VIII’ (loop 2), and motifs IX and X (loop 3), two Zinc-finger domains of METTL3, and the RGG motifs of METTL14. In the absence of a single experimental structure of all these domains, modeling of a DNA and RNA substrate in a full length METTL3-METTL14 heterodimer would be highly speculative. We will be performing molecular dynamics simulations and validate our results by extensive mutagenesis in the future. We did attempt to co-crystallize METTL3-METTL14 with several DNA and RNA sequences mentioned in this study, but unfortunately could not grow co-crystals yet.

The authors have tested the effect of structured RNAs on DNA methylation specifically, why not RNA substrates lacking structure. Is this a competition between substrates with different affinities and what could be the relevance in cellular processes specifically? Competition experiments haven't been performed here.

We did include the unstructured single stranded RNA such as rL3-5, rc-myc-p5 and observed consistent results i.e., the attenuated methylation of DNA substrate (d6T*). However, the level of inhibition of DNA methylation was more pronounced when structured RNAs lacking an ideal m6A target sequence e.g., rNEAT2 and rTCE23 were added to the reaction mixture. As shown in Figure 2e-g, we also performed competition experiments to confirm that the m6dA DNA methylation decreased in the presence of structured RNA. Moreover, data shown in Figure 2e suggests that the structured RNA (with high affinity to METTL3-METTl14) can attenuate m6dA DNA methylation in a dose dependent manner. These results indicate a competition between the RNA and the DNA substrate. The emerging evidence suggests that the recruitment of METTL3METTL14 and/or the presence of m6A methylated RNA at the site of DNA damage and repair (Zhang et al., Mol Cell 2020 Aug 6, 79, 425–442, PMID: 32615088; Xiang et al., Nature. 2017 Mar 23;543(7646):573-576). A more recent study confirmed the activity of METTL3-METTL14 on DNAs that contain various types of DNA damages (Yu et al., Nucleic Acid Res, 2021, 49 (20) 11629-42, PMID: 34086966). METTL3-METTL14 is also active on chromatin associated RNAs (Liu et al., Science 2020, 367, 580-586, PMID: 1949099). In the context of these studies, our results provide the first biochemical evidence for the interplay between METTL3-METTL14, DNA and RNA, and how the RNA and DNA methylation activities can be regulated in an environment enriched with all three molecules. Our work also presents strong rationale to decipher the physiological consequences of potentially new regulatory layer in the epitranscriptomic gene regulation.